# Regulation of Fibroblast Activation Protein by Transforming Growth Factor Beta-1 in Glioblastoma Microenvironment

**DOI:** 10.3390/ijms22031046

**Published:** 2021-01-21

**Authors:** Evzen Krepela, Zdislava Vanickova, Petr Hrabal, Michal Zubal, Barbora Chmielova, Eva Balaziova, Petr Vymola, Ivana Matrasova, Petr Busek, Aleksi Sedo

**Affiliations:** 1Laboratory of Cancer Cell Biology, Institute of Biochemistry and Experimental Oncology, First Faculty of Medicine, Charles University, 128 53 Prague 2, Czech Republic; evzen.krepela@lf1.cuni.cz (E.K.); zvani@lf1.cuni.cz (Z.V.); michal.zubal@lf1.cuni.cz (M.Z.); barbora.chmielova@lf1.cuni.cz (B.C.); eva.balaziova@lf1.cuni.cz (E.B.); petr.vymola@lf1.cuni.cz (P.V.); ivana.matrasova@lf1.cuni.cz (I.M.); 2Department of Pathology, Military University Hospital Prague, 169 02 Prague 6, Czech Republic; petr.hrabal@uvn.cz

**Keywords:** glioblastoma, tumor microenvironment, transforming growth factor beta, fibroblast activation protein, seprase, regulation of expression, Smad2, signaling

## Abstract

The proline-specific serine protease fibroblast activation protein (FAP) can participate in the progression of malignant tumors and represents a potential diagnostic and therapeutic target. Recently, we demonstrated an increased expression of FAP in glioblastomas, particularly those of the mesenchymal subtype. Factors controlling FAP expression in glioblastomas are unknown, but evidence suggests that transforming growth factor beta (TGFbeta) can trigger mesenchymal changes in these tumors. Here, we investigated whether TGFbeta promotes FAP expression in transformed and stromal cells constituting the glioblastoma microenvironment. We found that both FAP and TGFbeta-1 are upregulated in glioblastomas and display a significant positive correlation. We detected TGFbeta-1 immunopositivity broadly in glioblastoma tissues, including tumor parenchyma regions in the immediate vicinity of FAP-immunopositive perivascular stromal cells. Wedemonstrate for the first time that TGFbeta-1 induces expression of FAP in non-stem glioma cells, pericytes, and glioblastoma-derived endothelial and FAP^+^ mesenchymal cells, but not in glioma stem-like cells. In glioma cells, this effect is mediated by the TGFbeta type I receptor and canonical Smad signaling and involves activation of *FAP* gene transcription. We further present evidence of FAP regulation by TGFbeta-1 secreted by glioma cells. Our results provide insight into the previously unrecognized regulation of FAP expression by autocrine and paracrine TGFbeta-1 signaling in a broad spectrum of cell types present in the glioblastoma microenvironment.

## 1. Introduction

Glioblastomas are genetically, epigenetically, and phenotypically highly heterogeneous and invariably lethal brain tumors which devastate brain tissues via their infiltrative and invasive growth [1]. They consist of different, phenotypically varied cancer and stromal cell types, such as mesenchymal, endothelial, and immune cells, all of which participate in creating a dynamic tumor microenvironment [2]. Proteolytic networks acting in the glioblastoma microenvironment are of critical importance in the process of gliomagenesis [3]. In a previous study, we demonstrated increased expression of fibroblast activation protein (FAP) in a large proportion of human glioblastomas and its localization in both transformed and stromal cells [4]. FAP, also known as seprase, is a dimeric type II transmembrane glycoprotein which exhibits catalytic activity of a serine protease (EC 3.4.21.B28) that can act as a dipeptidyl peptidase or endopeptidase on oligo- and polypeptide substrates and preferentially cleaves post-proline peptide bonds [5,6,7]. FAP expression is very low in most healthy human tissues and cells, with the exception of multipotent bone marrow stromal cells [8], alpha cells of Langerhans islets [9], some dermal fibroblasts surrounding hair follicles [10], and skin melanocytes after ultraviolet irradiation [11]. Prominent FAP expression is, however, found in stromal cells including fibroblasts, fibroblast-like cells, and myofibroblasts at sites of active tissue remodeling, such as fibrosis, chronic inflammation, and healing wounds [5,12,13]. Moreover, FAP is frequently overexpressed in solid malignant tumors, where it is found in stromal cells, e.g., in cancer-associated fibroblasts and endothelial cells, and it is also present in premalignant and cancer cells [5,14,15,16,17,18]. Due to its frequent overexpression in the tumor microenvironment and its plasma membrane localization, FAP is considered a promising molecular target for the imaging and treatment of malignant tumors [19,20,21]. 

Current studies provide evidence that FAP can participate in complex processes that drive and maintain the progression of malignant tumors. These include cell proliferation, differentiation, signaling, adhesion, migration, extracellular matrix remodeling, epithelial–mesenchymal transition, angiogenesis, invasion, metastasis, and immunosuppression, although it seems that the role of FAP is highly context dependent and cell type specific [14,19,22,23]. Despite the expanding list of validated and candidate natural peptide and protein substrates of FAP [5,6,7], the biochemical functions of FAP in the abovementioned biological processes are not yet precisely understood. Alongside their proteolysis-dependent effects, membrane-bound FAP molecules can engage in binding interactions with other proteins, including those anchored in the plasma membrane and recruited in lipid rafts and invadopodia [24] or those present in the cell cytoplasm [25]. During its biogenetic trafficking, FAP may thus exhibit both proteolytic and non-proteolytic protein-recruiting and scaffolding functions. 

Which factors control FAP expression in glioblastomas is currently unknown. Among the various molecular subtypes of glioblastomas, we observed the highest FAP expression in the mesenchymal subtype [4]. This glioblastoma subtype is associated with a more aggressive phenotype and with signaling pathways involved in wound healing and inflammation [26]. An important mediator of mesenchymal changes in glioblastomas is the transforming growth factor beta (TGFbeta) [27,28], a pleiotropic cytokine which exists in three isoforms (TGFbeta-1, -2, and -3), and by binding to TGFbeta receptors, triggers both Smad and non-Smad TGFbeta-activated intracellular signaling pathways [29,30]. Depending on the type and differentiation status of the cell as well as activities of other signaling pathways, TGFbeta induces epithelial–mesenchymal transition, activates angiogenesis, dampens immune responses, and promotes tumor invasiveness [31,32]. Since it has been shown that in some, but not all cell types, the *FAP* gene is a transcriptional target for TGFbeta signaling [23], we studied several cell types isolated from or serving as model systems for the complex glioblastoma microenvironment to assess whether TGFbeta participates in the upregulation of *FAP* gene expression in particular subpopulations of transformed and stromal cells typically present in glioblastomas. 

## 2. Results

### 2.1. Expression of FAP and TGFbeta Isoforms in Human Glioblastomas

The levels of FAP protein (Figure 1A, left panel) and FAP enzymatic activity (Figure 1A, right panel) were substantially higher in glioblastomas than in non-tumorous pharmacoresistant epilepsy (PRE) brain tissues. In glioblastomas, the enzymatic activity of FAP displayed statistically significant positive correlation with FAP protein concentration (Figure 1D). The strength of this correlation was moderate, which may be due to differences in the relative concentrations of active FAP molecules [33]. ELISAs specific to the individual TGFbeta protein isoforms revealed that in glioblastomas, TGFbeta-1 was the most abundantly expressed TGFbeta isoform (Figure 1B). Further analysis revealed that the concentration of TGFbeta-1 protein was substantially higher in glioblastomas than in the PRE brain tissues (Figure 1C). In glioblastomas, FAP and TGFbeta-1 protein concentrations showed a weak but statistically significant positive correlation (Figure 1E). 

We previously reported that FAP was most prominently upregulated in the mesenchymal subtype of glioblastoma based on the data from 173 glioblastomas in The Cancer Genome Atlas (TCGA) database [4]. We now extended the analysis of the TCGA data to 505 primary glioblastomas and analyzed the expression of FAP and individual TGFbeta isoforms. Concordantly with the FAP protein data reported in the present study, FAP mRNA was upregulated in glioblastomas compared to control brain tissues (Figure 2). Similarly, transcripts encoding TGFbeta isoforms (TGFB 1, 2, 3) were upregulated in glioblastomas compared to control brain tissues (Figure 2) as was also reported in previous studies [34]. Expression of FAP and all TGFbeta isoforms was highest in the mesenchymal subtype glioblastomas (Figure 2). FAP expression correlated with TGFB1 and TGFB3, but not TGFB2 in all glioblastomas and in the mesenchymal subtype (Figure 2). Collectively, these data suggest that TGFbeta-1 may contribute to the regulation of FAP expression in the glioblastoma microenvironment. The TCGA data further suggest that this may be most pronounced in the mesenchymal subtype of glioblastoma.

### 2.2. Immunohistochemical Localization of FAP and TGFbeta-1 in Human Glioblastomas

Using chromogenic and fluorescence immunohistochemistry, we detected FAP immunopositivity in glioblastomas in several cell subpopulations including perivascular stromal cells, some intraparenchymal cancer cells in approximately 25% of glioblastomas and, albeit infrequently, endothelial cells of some blood vessels (Figure 3A–D,F). In contrast to glioblastomas, PRE brain tissues did not exhibit FAP immunopositivity (Figure 3E). By means of fluorescence immunohistochemistry and confocal microscopy, we detected TGFbeta-1 immunopositivity broadly in glioblastoma tissues including regions in the immediate vicinity of FAP-immunopositive perivascular stromal cells (Figure 3F).

### 2.3. Upregulation of FAP Enzymatic Activity and FAP Protein Induced by Recombinant TGFbeta-1 in Different Cell Types Present in the Glioblastoma Microenvironment

To determine whether TGFbeta-1 may contribute to FAP upregulation in glioblastomas, we cultured several different cell types representing cellular components of glioblastoma microenvironment, with and without human recombinant TGFbeta-1 for 72 h, and measured FAP enzymatic activity. Permanent human glioma cell lines U87, U251 and U118 displayed detectable baseline FAP enzymatic activity and FAP protein which dose-dependently increased when the cells were cultured in the presence of TGFbeta-1 at concentrations ranging from 0.2 ng⋅mL^−1^ to 10 ng⋅mL^−1^ (Figure 4A and Figure 5). A considerable TGFbeta-1-mediated upregulation of FAP enzymatic activity was also observed in four out of five tested non-stem glioma cell cultures (Figure 4B, left part), while TGFbeta-1-treated glioma stem-like cell cultures showed either no change or negligible increase in their extremely low baseline FAP enzymatic activity (Figure 4B, right part). Human brain vascular pericytes (HBVP) and glioblastoma-derived FAP^+^ mesenchymal (pFAP) cell cultures displayed a marked upregulation of their FAP enzymatic activity after treatment with TGFbeta-1 (Figure 4C). Glioblastoma-derived endothelial cell (pEC) cultures showed a highly variable level of baseline FAP enzymatic activity but all exhibited a significantly upregulated FAP enzymatic activity after treatment with TGFbeta-1 (Figure 4D). In contrast to the pEC cultures, human umbilical vein endothelial cells (HUVEC) showed no TGFbeta-1-mediated increase in their extremely low baseline FAP enzymatic activity (Figure 4D). Using ELISA, we confirmed FAP upregulation on the protein level in permanent glioma cell lines and endothelial cells (Figure 5). In all investigated cell types that responded to TGFbeta-1 by FAP activity and FAP protein upregulation, the maximum response was reached at 2 ng⋅mL^−1^ of TGFbeta-1 with no further increase for 10 ng⋅mL^−1^ (Figure 4 and Figure 5). To the best of our knowledge, there is no evidence that the catalytic activity of FAP is regulated by an endogenous biological inhibitor or activator. In cells expressing FAP, the levels of FAP enzymatic activity and FAP protein concentration are upregulated in parallel after treatment with different concentrations of recombinant TGFbeta-1 (Figure 4 and Figure 5). This results in a strong positive Pearson correlation between FAP protein and FAP activity levels, as shown for human glioma cell lines U87 and U251 and for endothelial cells pEC 54A (Figure 5). Therefore, the quantification of FAP enzymatic activity is a reliable measure of *FAP* gene expression level in various cell types.

### 2.4. Characterization of the TGFbeta-1-Mediated Upregulation of FAP Enzymatic Activity and FAP Protein in Human Glioma Cells

Besides being dose-dependent (Figure 4 and Figure 5), TGFbeta-1-mediated upregulation of FAP protein concentration and FAP enzymatic activity in human glioma cells was also time-dependent (Figure 6B,C). Moreover, the TGFbeta-1-treated U87 glioma cells also displayed a time-dependent upregulation of FAP mRNA (Figure 6A). The TGFbeta-1-induced increase in FAP mRNA expression preceded that of FAP protein by at least 12 h (Figure 6A,B), which indicates that TGFbeta-1 regulates FAP expression on the transcriptional level. Interestingly, we also observed an upregulation of FAP mRNA, FAP protein, and FAP enzymatic activity over time in our control cell cultures (Figure 6A–C,F). 

The synthetic small-molecule inhibitors of TGFbeta type I receptor (ALK-5 kinase) A8301 and A7701 [35] not only completely blocked the exogenous recombinant TGFbeta-1-mediated upregulation of FAP protein and FAP enzymatic activity in U87 glioma cells (Figure 6B,D), but also led to a significant decrease in the baseline level of FAP protein and FAP enzymatic activity in the cultured cells (Figure 6D). Since this A8301- and A7701-sensitive fraction of baseline FAP protein and FAP enzymatic activity could be due to upregulation via autocrine TGFbeta-1 signaling, we investigated the secretion of endogenous TGFbeta-1 in U87 and U251 glioma cells and in human brain vascular pericytes (HBVP). By means of ELISA, we demonstrated that all three studied cell lines secreted TGFbeta-1 into serum-free media, albeit in different quantities (Figure 6E). The baseline level of FAP enzymatic activity in cultured U87 cells significantly increased over time and reached a plateau after several cultivation days (Figure 6F). However, it was markedly reduced (by 32–42%) when U87 cells were cultured in the presence of A8301 and A7701 (Figure 6F). The inhibitor A7701 is a direct synthetic precursor of A8301, which exerts about twice as much a potent inhibitory effect (in terms of ID_50_) against the TGFbeta type I receptor as compared to A7701 [35]. Nevertheless, at a treatment concentration of 1 µM, both inhibitors completely prevented the TGFbeta-1-induced upregulation of FAP protein and FAP enzymatic activity in U87 glioma cells (Figure 6B,D). 

To assess whether TGFbeta-1 signaling leads selectively to FAP upregulation, we evaluated the effect of TGFbeta-1 on the expression of dipeptidyl peptidase-4 (DPP4). This proline-specific protease is the closest FAP homologue [36] and is co-expressed with FAP in some cell types including glioma cells [9,37]. Nevertheless, in U87 glioma cells cultured with or without recombinant TGFbeta-1 for 72 h, the level of DPP4 protein and DPP4 enzymatic activity remained unchanged (Figure 7). 

### 2.5. TGFbeta-1-Induced Phosphorylation of Smad2 in FAP-Upregulating and FAP-Non-Upregulating Human Glioma Cells

Since the U87 glioma cells and the AZVU001A glioma stem-like cells were sensitive and fully resistant, respectively, to TGFbeta-1-mediated upregulation of FAP enzymatic activity (Figure 4A,B), we wondered whether this difference may be due to the absence of activation of the canonical TGFbeta receptor/Smad signaling pathway in the glioma stem-like cells. Using Western blot analysis with specific anti-pSmad2 antibodies, we demonstrated that both the FAP-upregulating U87 glioma cells and FAP-non-upregulating AZVU001A glioma stem-like cells were capable of rapidly phosphorylating the Smad2 protein when exposed to TGFbeta-1 (Figure 8A,B). TGFbeta-1-induced phosphorylation of Smad2 was completely inhibited when the cells were co-treated with TGFbeta-1 and the inhibitor A7701 (Figure 8A,B; for the full length immunoblots see the Appendix A).

## 3. Discussion

Proteases contribute to cancer pathogenesis and represent potential diagnostic and therapeutic targets. In glioblastoma, TGFbeta is produced by glioma cells, microglia, and astrocytes [38,39,40]. TGFbeta signaling through phosphorylated Smad2 regulates expression of various TGFbeta-response genes [27,41,42,43] and leads—in addition to various other effects—to dysregulation of proteolytic networks [3]. Our previous work suggested increased expression of FAP, a potential TGFbeta target gene, in glioblastomas [4]. It this work we investigated the effect of TGFbeta signaling on FAP expression in multiple cell types present in glioblastoma microenvironment. 

We show that all three TGFbeta protein isoforms (-1, -2, and -3) are expressed in glioblastomas, with the TGFbeta-1 isoform being the most abundantly produced. This is in line with recent findings in glioblastomas obtained by methods other than ELISA [34,41]. We used biochemical and immunohistochemical methods to show that the expression of both FAP and TGFbeta-1 was upregulated in glioblastomas as compared to non-tumorous brain tissues. Interestingly, the FAP-immunopositive cells within the glioblastoma microenvironment were localized in the immediate vicinity of the TGFbeta-1 immunopositive stromal and cancer cells. These observations in conjunction with the positive correlation between FAP and TGFbeta-1 protein levels in glioblastomas suggested that TGFbeta-1 signaling may be involved in the regulation of FAP expression by targeting multiple cell types within the glioblastoma microenvironment. To test this hypothesis, we carried out in vitro experiments, in which, several cell types representing the cellular components of the glioblastoma microenvironment were exposed to human recombinant TGFbeta-1. These experiments revealed that TGFbeta-1 upregulates the expression of FAP in human permanent and non-stem glioma cell cultures, brain vascular pericytes, and glioblastoma-derived endothelial cells and FAP^+^ mesenchymal cells. 

It was previously reported that human glioma cell lines secrete both TGFbeta-1 and TGFbeta-2 proteins [40]. Both secreted TGFbeta isoforms occur as a mixture of a latent precursor and a biologically active form, and the proteolytic conversion of the former to the latter is catalyzed by a furin-like protease [40]. In cultured U87 and U251 glioma cell lines, we observed an accumulation of the secreted endogenous TGFbeta-1 in the culture medium and a parallel increase in the baseline FAP protein and FAP activity levels in the cells. These observations together with the ability of recombinant TGFbeta-1 to upregulate FAP in the cells pointed to the possible regulation of FAP level through the endogenous TGFbeta-1/TGFbeta receptor signaling axis. To test this, we exposed the cultured glioma cells to specific TGFbeta type I receptor inhibitors A8301 and A7701 [35]. This resulted in a significant decrease in FAP protein and FAP activity levels in the inhibitor-treated glioma cells providing evidence that autocrine/paracrine TGFbeta-1 signaling can contribute to the regulation of baseline FAP levels. These results, together with the ability of human brain vascular pericytes to secrete TGFbeta-1, further indicate that the autocrine/paracrine TGFbeta-1 signaling can contribute to the regulation of FAP level in multiple cell types within the glioblastoma microenvironment. 

Several recent studies demonstrated that the specific chemical inhibitors of the TGFbeta type I receptor, including A8301 and A7701 used in our experiments, effectively suppress the canonical Smad-dependent TGFbeta signaling pathway, but they do not interfere with the TGFbeta-triggered non-Smad signaling pathways including those involving MAP kinases [35,44,45,46,47]. Thus, the complete suppression of the TGFbeta-1-induced upregulation of FAP and the partial suppression of the baseline level of FAP we observed after treatment with A8301 and A7701, are caused by the specific blocking of the canonical Smad-dependent TGFbeta signaling pathway. This is further supported by the concurrent lack of Smad2 phosphorylation in U87 cells which were co-treated with TGFbeta-1 and A7701. These data together provide evidence that the canonical Smad-dependent TGFbeta signaling axis accounts for the TGFbeta-1-mediated regulation of FAP expression in glioblastoma cells. 

It has previously been demonstrated that TGFbeta-1 can trigger the upregulation of *FAP* gene expression in certain human melanoma cell lines via a signaling pathway that involves the activation of TGFbeta receptors and formation of phosphorylated Smad (pSmad) complexes which directly bind to the *FAP* promoter and activate *FAP* gene transcription [23]. Besides this mechanism, other transcription factors, including EGR-1 [48], Twist [49] and Snail [50], can likewise directly bind to the *FAP* promoter and trigger the transcription of the *FAP* gene. Interestingly, in some cell types, the genes that encode EGR-1, Twist, and Snail can be transcriptionally activated by the TGFbeta-induced pSmad complexes [51,52,53]. Nevertheless, there is evidence that pSmad-mediated TGFbeta-1 signaling does not upregulate the levels of Snail and Twist proteins in human glioma cells [54]. This supports the conclusion that TGFbeta signaling induces direct activation of *FAP* gene expression by pSmad complexes in glioma cells. Thus, mesenchymal transcription factors such as Twist [49] and the autocrine and paracrine TGFbeta-1 signaling described in the current work contribute to the regulation of FAP expression in glioma cells.

The biochemical basis of the very low baseline FAP levels and absent or negligible upregulation of FAP after treatment with recombinant TGFbeta-1 in glioma stem-like cells and human umbilical vein endothelial cells (HUVEC) is, at present, unknown. We assume that the protein-binding partners of pSmad complexes, which can function as switching contextual determinants [32,55], modify the functionality of the TGFbeta-1 signaling pathway controlling *FAP* gene expression in these cells. Further studies are warranted to identify the molecular mechanism by which cells avoid the TGFbeta-1-mediated upregulation of FAP expression, as they could contribute to the identification of factors maintaining the extremely low FAP expression in the majority of healthy tissues.

The functions of FAP in cancer are incompletely understood and seem very probably different in different cell types. Its role in individual cell subpopulations in the glioblastoma microenvironment remains to be established. Nevertheless, approaches utilizing FAP expression for tumor imaging, including imaging of glioblastomas, and targeted delivery of anticancer therapeutics have been recently described [19,20,21]. Considering the important role of TGFbeta-1 signaling in glioblastoma progression, the TGFbeta-induced expression of FAP opens an interesting possibility of evaluating these approaches for parallel therapeutic targeting of several cell subpopulations in glioblastoma. 

## 4. Materials and Methods 

### 4.1. Patients and Tissue Samples

Tissue samples were obtained from surgically treated patients (Table 1) with glioblastoma multiforme (GBM, WHO grade IV glioma, n = 76) and pharmacoresistant epilepsy (PRE; n = 10; controls) and were stored at −78 °C until preparation of lysates. The tumors were graded according to the current WHO classification [56]. The study was conducted in accordance with the Helsinki Declaration and was approved by the Ethics Committees of the Na Homolce Hospital, the Military University Hospital, both Prague, Czech Republic, and the Masaryk Hospital, Usti nad Labem, Czech Republic (study approval numbers 108-39/4-2014-UVN and 7/8/2014-25, approved on 28 July 2014 and 21 July 2014, respectively). Signed informed consent was obtained from all patients entering the study. 

### 4.2. The Cancer Genome Atlas (TCGA) Database Data

Expression data and information on the molecular subtype of primary glioblastomas (n = 505) and non-tumorous brain tissues (n = 10) from the TCGA GBM (HG-UG133A platform) were downloaded from GlioVis data portal [57] on 29 February 2020. 

### 4.3. Immunohistochemistry and Confocal Microscopy

For FAP immunostaining in 4 µm paraffin sections, we used a primary rabbit monoclonal antibody (Ab207178 (clone EPR20021), Abcam, Cambridge, UK; 1:250, at room temperature for 20 min) and an automated Bond Stainer (Leica, Buffalo Grove, IL,USA). Antigen retrieval was performed with an EDTA-based pH 9.0 epitope retrieval solution (BOND Epitope Retrieval Solution 2) for 20 min and Bond Polymer Refine Detection (Leica) was used to visualize the primary antibody and haematoxylin counterstaining. Images were captured by an experienced pathologist (P.H.) on an Axioskop 2 mot plus microscope using the Axiocam ICc1 camera (Zeiss, Oberkochen, Germany). Sequential double immunofluorescence labelling of TGFbeta-1 and FAP was performed in 10 µm frozen sections with minor modifications of a previously described protocol [4,9]. The sections were fixed with chilled aceton:methanol 1:1 at −20 °C for 5 min, blocked with 10% fetal calf serum plus 1% bovine serum albumin in TBS, and incubated with primary anti-TGFbeta-1 antibody (ab92486, Abcam; 1:400, overnight at 4 °C). After washing away non-bound antibodies, the slides were incubated with the primary anti-FAP antibody (D8; Vitatex, Stony Brook, NY, USA; 1:200, 1 h at room temperature). After washing, slides were incubated with the corresponding Alexa Fluor 488- and 546-conjugated secondary antibodies (Thermo Fisher Scientific, Waltham, MA, USA; 1:500, 1 h at room temperature). 400 µM ToPro (Thermo Fisher Scientific) or 50 ng⋅mL^−1^ Hoechst 33,258 (Sigma-Aldrich, St. Louis, MO, USA) were used for counterstaining of cell nuclei. Immunostained sections were viewed and photographed using Olympus IX 81 confocal microscope (FluoView 300, Olympus, Shinjuku, Japan).

### 4.4. Cell Culture and Experimental Treatment of Cells 

Permanent human glioma cell lines U87, U251 and U118 were purchased from CLS Cell Lines Service GmbH, Eppelheim, Germany. The cells were grown in Dulbecco’s Modified Eagle’s Medium (DMEM; Sigma-Aldrich, Cat. No. D5796), supplemented with 10% of fetal calf serum (FCS), at 37 °C in a humidified atmosphere of 5% CO_2_ in air. 

Cultures of non-stem and stem-like cells from glioblastomas were derived, cultured and characterized as we described previously [58]. Briefly, after mechanical and enzymatic (Papain Dissociation System, Worthington, Lakewood, NJ, USA) dissociation according to manufacturer’s recommendations, cells were grown in 10% FCS-containing media (non-stem cell cultures), or after forming gliomaspheres in serum-free media (stem-like cell cultures, DMEM/F12 with bFGF 20 ng⋅mL^−1^ and EGF 20 ng⋅mL^−1^, both PeproTech, London, UK), B27-supplement 1:50 (Thermo Fisher Scientific), 1% Glutamax (Thermo Fisher Scientific), 100 U⋅mL^−1^ penicillin and 100 μg⋅mL^−1^ streptomycin) propagated on Geltrex-coated (Thermo Fisher Scientific) plastic. The cells were grown at 37 °C in a humidified atmosphere of 5% of CO_2_ in air. The three studied glioma stem-like cell cultures were all tumorigenic in immunodeficient mice forming highly infiltrative tumors. These cells cultures expressed in variable extent the stem cell markers CD133 (anti-AC133-APC, Miltenyi Biotec, Bergisch Gladbach, Germany) and Sox2 (anti-Sox2-PerCP, IC2018C, R&D Systems, Abingdon, UK) as determined by flow cytometry [58]. They also differentiated to glial fibrillary acidic protein (GFAP) and beta-III tubulin positive cells when exposed to 10% FCS containing medium [58], except for GSC247 cells. Of the five studied non-stem glioma cell cultures, four were tumorigenic in immunodeficient mice.

Human brain vascular pericytes (HBVP, ScienCell Research Laboratories Carlsbad, CA, USA) were cultured according to the manufacturer’s recommendations on polylysine-coated (0.01%, Sigma-Aldrich) plastic in the recommended Pericyte Medium (ScienCell, Cat. No 1201) supplemented with pericyte growth supplement (PGS), 2% FCS and 100 U⋅mL^−1^ penicillin and 100 μg⋅mL^−1^ streptomycin. Human umbilical vein endothelial cells (HUVEC) were obtained from Thermo Fisher Scientific and cultured according to the manufacturer’s recommendations in M200 (Cascade Biologics Thermo Fisher Scientific) supplemented with 2% Large Vessel Endothelial Supplement (LVES, Thermo Fisher Scientific). 

Endothelial cell cultures (pEC) were derived from fresh glioblastoma tissue. Tumor tissue was cut in small pieces and digested with TrypLE Select (Thermo Fisher Scientific) at 37 °C for 20 min. The resulting cell suspension was used for direct magnetic-activated cell sorting (MACS) using a CD105 antibody (M3527, Agilent Dako, Santa Clara, CA, USA 0.15 µg of the antibody per 100 µL of 10^7^ cells per mL at 4 °C for 8 min) and Sheep-Anti Mouse IgG Dynabeads (Cat. No. 11031, Thermo Fisher Scientific, 0.15 µL per 100 µL of 10^7^ cells per mL) according to the manufacturer’s recommendations. The positive fraction was cultured as described by Miebach et al. [59] and Charalambous et al. [60] on fibronectin-coated (3 µg⋅cm^−2^, Sigma-Aldrich) plastic in an initiation medium containing RPMI, 10% FCS, 10% Nu serum (BD Bioscience, San Jose, CA, USA), 1 mM HEPES (Sigma-Aldrich), 300 UI Heparin (Zentiva), 1% penicillin/streptomycin (Sigma-Aldrich), and 3 μg⋅mL^−1^ endothelial cell growth supplement (ECGS, BD Bioscience). After two weeks, 10% Nu Serum was omitted from culture media and the ECGS concentration was increased to 30 µg⋅mL^−1^. Alternatively, the mechanically and enzymatically dissociated tissue fragments were placed on fibronectin-coated plastic in the initiation medium and MACS was performed after harvesting the emigrated cells, usually after 7 days. Thereafter, the cells were cultured as described above. A similar procedure was used to isolate FAP^+^ mesenchymal cells (pFAP), except that an anti-FAP F11-24 antibody (Santa Cruz Biotechnology, Dallas, Texas, USA; 0.1 µg of the antibody per 100 µL of 10^7^ cells per mL at 4 °C for 8 min) was used for MACS and cells were grown in pericyte growth medium (Pericyte medium, ScienCell, Cat. No. 1201, supplemented with 2% FCS, PGS, 100 U⋅mL^−1^ penicillin and 100 μg⋅mL^−1^ streptomycin). The isolated endothelial cells expressed the endothelial cell marker von Willebrand factor (vWF, in 100% of cells) and did not express astrocytic (GFAP) or mesenchymal (clone TE7, platelet derived growth factor receptor—PDGFR) markers. The pFAP cell cultures expressed mesenchymal markers (FAP in approximately 70% of cells, TE7 in approximately 90% of cells, PDGFRbeta in approximately 90% of cells) and were GFAP and vWF negative as determined by immunocytochemistry using specific primary antibodies (Table 2)

During experiments, permanent glioblastoma cell lines, glioma non-stem and stem-like cell cultures, human brain vascular pericytes (HBVP), glioblastoma-derived FAP^+^ mesenchymal cells (pFAP), glioblastoma-derived endothelial cells (pEC), and human umbilical vein endothelial cells (HUVEC) were cultured in the appropriate media (see above) in 6-well culture plates (Nunc, Roskilde, Denmark) at 37 °C in a humidified atmosphere of 5% of CO_2_ in air. After 72 h, the culture media were exchanged for fresh media with or without the tested substances. Human recombinant transforming growth factor beta-1 (TGFbeta-1; Cat. No. 100-21C, PeproTech, Cranbury, NJ, USA), 10 µg⋅mL^−1^ in PBS, pH 7.2, containing 0.1% of bovine serum albumin (BSA), was added to the culture medium to achieve the treatment concentration of 0.2, 2 and 10 ng⋅mL^−1^. In some experiments, potent inhibitors of the TGFbeta type I receptor (activin-like kinase 5/ALK-5 kinase) A8301 [35] and A7701 [35] (both from Axon Medchem, Groningen, The Netherlands) were used. The inhibitors were dissolved at a concentration of 10 mM in dimethyl sulfoxide and were further diluted to a treatment concentration of 1 µM in the culture medium either alone or with recombinant TGFbeta-1. Cells were then grown under the described conditions for the required period of time and were harvested and processed as described below. 

### 4.5. Preparation of Tissue and Cell Lysates and Cell Conditioned Media

Tissue samples were homogenized with Ultra-Turrax homogenizer (T10; IKA, Königswinter, Germany) on ice in a homogenization buffer containing 2 mM Na_2_HPO_4_, 0.6 mM KH_2_PO_4_ and 22.4 mM NaCl, pH 6.0. The 15% homogenates were mixed in 1:1 volume parts with a lysis buffer (10 mM Tris-HCl, pH 7.5, containing 1% Triton X-100, 0.1% SDS, 100 mM NaCl, 1 mM EDTA, 1 mM EGTA, 10% glycerol) and a mix of protease and phosphatase inhibitors including 25 μM pepstatin A, 200 μM AEBSF, 50 μM E-64, 5 mM NaF, and 1 mM Na_3_VO_4_. The suspensions were rotation-mixed at 4 °C for 30 min and then cleared by centrifugation at 22,000× *g* and 4 °C for 30 min. The cleared supernatants, i.e., the tissue lysates, were collected and stored in aliquots at −75 °C until analysis. Cell lysates were prepared as follows: cells adherently growing in 6-well culture plates (Nunc) were washed twice with ice-cold PBS buffer pH 7.4 and scraped off in 200 μL of the lysis buffer on ice. Cell lysates were cleared by centrifugation at 22,000× *g* and 4 °C for 30 min and the supernatants were stored in aliquots as described above. 

To obtain serum-free conditioned media from U87 and U251 cells, the cells were seeded in 6-well Nunc culture plates and grown in DMEM supplemented with 10% FCS, at 37 °C in a humidified atmosphere of 5% CO_2_ in air to reach confluence. The medium was then removed, cells were washed twice with PBS buffer pH 7.4, and cultivated in 2 mL of fresh serum-free DMEM per well for additional 72 h. The sampling was done at 2, 4, 6, 12, 24, 48, and 72 h of cultivation and involved the withdrawal of 200 μL of the conditioned culture medium and immediate volume replenishment with 200 μL of fresh, serum-free DMEM. To obtain the conditioned medium from HBVP cells, the cells were seeded in 6-well Nunc culture plates and grown in a complete Pericyte Medium (see above) to reach confluence. The medium was then removed, the cells washed with PBS buffer pH 7.4, and cultivated in 2 mL of fresh, serum-free Pericyte Medium. Sampling of the conditioned medium was done as described above except that fresh serum-free Pericyte Medium was used for volume replenishment. The collected conditioned media were centrifuged at 22,000× *g* and 4 °C for 30 min to remove detached cells and cell debris, and the supernatants were stored in aliquots at −75 °C until analysis. 

### 4.6. Determination of Total Protein

Total protein concentration in tissue and cell lysates was determined by the Lowry method [61] using BSA as a standard.

### 4.7. Determination of Enzyme Activity

Enzymatic activity of FAP was measured in black, flat bottom 96-well plates (Corning Costar, Tewksbury, MA, USA) using a kinetic assay with 150 µM of *N*-(quinoline-4-carbonyl)-*D*-Ala-*L*-Pro-7-amido-4-methyl-coumarin as a fluorogenic FAP-specific substrate [62] prepared by standard Boc peptide chemistry [63]. FAP assays were carried out at 37 °C in a total reaction volume of 100 µL in a PBS buffer containing 8 mM NaH_2_PO_4_, 42 mM Na_2_HPO_4_ and 150 mM NaCl, pH 7.4. Fluorescence of the enzymatically released 7-amino-4-methylcoumarin (AMC) was measured on a microplate fluorimeter Infinite M1000 (Tecan, Grödig, Austria) using excitation and emission wavelengths/slits of 380/5 nm and 460/5 nm, respectively. During the assays, less than 3% of the initial FAP substrate concentration was cleaved. The FAP substrate on its own was stable in the PBS buffer pH 7.4. Measurements were done in triplicate and were calibrated with several concentrations of AMC in the assay buffer. 

The enzymatic activity of DPP4 was measured in white, flat bottom 96-well plates (Nunc) at 37 °C using a kinetic assay with 50 μM of Gly-*L*-Pro-AMC (Bachem, Bubendorf, Switzerland) as a fluorogenic substrate in PBS buffer pH 7.4. In parallel assays, a highly selective DPP4 inhibitor sitagliptin (Biovision, Milpitas, CA, USA) was used at 300 nM to exclude the activity of other DPP4-related proteases cleaving the substrate, as described previously [64]. 

### 4.8. ELISA Assays

TGFbeta protein isoforms TGFbeta-1, TGFbeta-2 and TGFbeta-3, and proline-specific protease proteins FAP and DPP4 were assayed in tissue and cell lysates and serum-free conditioned media by DuoSet ELISA kits (R&D Systems) including DuoSet TGF beta-1 (Cat. No. DY240), DuoSet TGF beta-2 (Cat. No. DY302), DuoSet TGF beta-3 (Cat. No. DY243), DuoSet FAP (Cat. No. DY3715) and DuoSet DPP-IV (Cat. No. DY1180), respectively, in accordance with the manufacturer’s recommendations.

### 4.9. Western Blot Analysis 

The Western blot procedure was performed in a similar way to that described previously [65] with some modifications specified below. The lysate samples, 35 μg of total protein per one gel lane, were electrophoresed under denaturing and reducing conditions in discontinuous (5% stacking, 10% resolving) 1.5 mm polyacrylamide gels. The samples were mixed with a 5× sample buffer containing 10 *w*/*v*% SDS, 50 *w*/*v*% glycerol, 250 mM Tris/HCl, 0.05% Serva Blue G and 500 mM dithiothreitol, pH 7.40. Electrophoretic separation was carried out in an electrode buffer containing 25 mM Tris, 192 mM glycine and 0.1% SDS, pH 8.3, at constant voltage (60 V for 30 min followed by 140 V for 90 min). After separation, the gels were equilibrated in a transfer buffer containing 48 mM Tris, 39 mM glycine, pH 9.2, and 20% methanol. Proteins were transferred onto a polyvinylidene difluoride (PVDF) membrane using a semidry blotting system (Bio-Rad, Hercules, CA, USA). PVDF membranes with the transferred proteins were thoroughly rinsed in a buffer (TTSB) containing 0.05% Tween 20, 100 mM Tris/HCl and 154 mM NaCl, pH 7.5, and blocked in a 5% BSA in TTSB prior to incubation with a 1000× diluted primary antibody Rabbit Anti-Phospho Smad2 (Ser465/467) (Sigma-Aldrich; Cat. No. AB3849-I) at 4 °C overnight. After extensive washing in TTSB, the membranes were incubated with a 25,000× diluted Donkey Anti-Rabbit IgG-Horseradish Peroxidase-Conjugated secondary antibody (GE Healthcare, Little Chalfont, UK; Cat. No. NA934) at room temperature for 60 min. Immunoblots were washed with TTSB and developed using Luminata Forte (Sigma-Aldrich). Chemiluminiscence signal was detected and captured on an imaging ChemiDoc System (BioRad). Subsequently, the membranes were washed with TTSB and probed overnight at 4 °C for glyceraldehyde 3-phosphate dehydrogenase (GAPDH), which served as a protein-loading control, using a 1000× diluted Rabbit Anti-GAPDH (Sigma-Aldrich; Cat. No. 2459656). After extensive washing with TTSB, the membranes were incubated with 20,000× diluted secondary Donkey Anti-rabbit IgG-horseradish peroxidase-conjugated antibody. The immunoblots were washed and developed with Luminata Forte. The signal was detected and captured on the ChemiDoc System. 

### 4.10. Isolation of Total RNA and Real Time RT-PCR 

The isolation and fluorometric quantification of total RNA from cultured human glioma cells and real time RT-PCR were carried out according to the previously reported procedures [66] using several important modifications in the real time RT-PCR method as described below. The forward and reverse primers and TaqMan probes for the quantification of expression of human FAP and β-actin (ACTB) mRNAs (Table 3) were designed using the Primer Express software (Thermo Fisher Scientific) and custom-synthesized at Thermo Fisher Scientific. 

The expression of human FAP and ACTB mRNAs was quantitated by a two-step, real-time RT-PCR assay as follows. In the first reverse transcription (RT) step, 3 μg of total RNA was reverse-transcribed in a total volume of 30 μL of 50 mM Tris/acetate buffer, pH 8.3, containing 75 mM of potassium acetate, 3 mM of MgCl_2_, 2.5 μM of Oligo (dT)_20_ Primer (Thermo Fisher Scientific), 500 μM of each dGTP, dCTP, dATP, and dTTP, 5 mM dithiothreitol, 60 units of RNase inhibitor RNAseOUT (Thermo Fisher Scientific) and 300 units of SuperScript III Reverse Transcriptase (Thermo Fisher Scientific). RT reactions were carried out at 55 °C for 30 min and were terminated by heating at 70 °C for 15 min. The resulting RT mixes were stored at −25 °C until PCR analysis. The subsequent PCR step was carried out in a total volume of 25 μL of 20 mM Tris/HCl buffer, pH 8.4, containing 50 mM of KCl, 4 mM of MgCl_2_, 0.43 mM of each dGTP, dCTP, dATP, and dTTP, 200 nM of each forward and reverse gene-specific primer and TaqMan probe (Table 3), 2.5 unit of Platinum Taq DNA polymerase (Thermo Fisher Scientific), and an aliquot of the RT mix, representing a PCR input equivalent of 200 ng of total RNA. PCR amplification included a hot start at 95 °C for 3 min and 40 cycles of denaturation at 95 °C for 15 sec and annealing/extension at 58 °C for 1 min. The real-time PCR assays were run in triplicate on the QuantStudio 12K Flex instrument (Thermo Fisher Scientific) operated from within the QuantStudio 12K Flex Software (Thermo Fisher Scientific). Threshold cycle (C_T_) values of amplification reactions, represented by the plots of the background subtracted fluorescence intensity (ΔFI) of the reporter dye (6-FAM) against the PCR cycle number, were determined using the QuantStudio 12K Flex Software. 

### 4.11. Statistical Analysis

Statistical calculations were performed with SigmaStat (Systat Software, San Jose, CA, USA) and Stat200 (Biosoft, Cambridge, UK). A two-sided *p* < 0.05 was considered statistically significant. Statistical difference of PCR results was calculated from linearized ΔC_T_ data (i.e., 2^−ΔC^_T_) after normalizing the expression of FAP mRNA (target transcript) on the expression of ACTB mRNA (reference transcript) [66]. 

## 5. Conclusions

Our present work demonstrates for the first time that FAP expression positively correlates with TGFbeta expression in glioblastomas and can be differentially regulated via autocrine and paracrine TGFbeta-1 signaling in non-stem glioma cells, pericytes, glioblastoma-derived endothelial and FAP^+^ mesenchymal cells, but not in glioma stem-like cells. In glioma cells, this regulation is mediated by TGFbeta type I receptor and canonical SMAD and involves the activation of *FAP* gene transcription. 

## Figures and Tables

**Figure 1 ijms-22-01046-f001:**
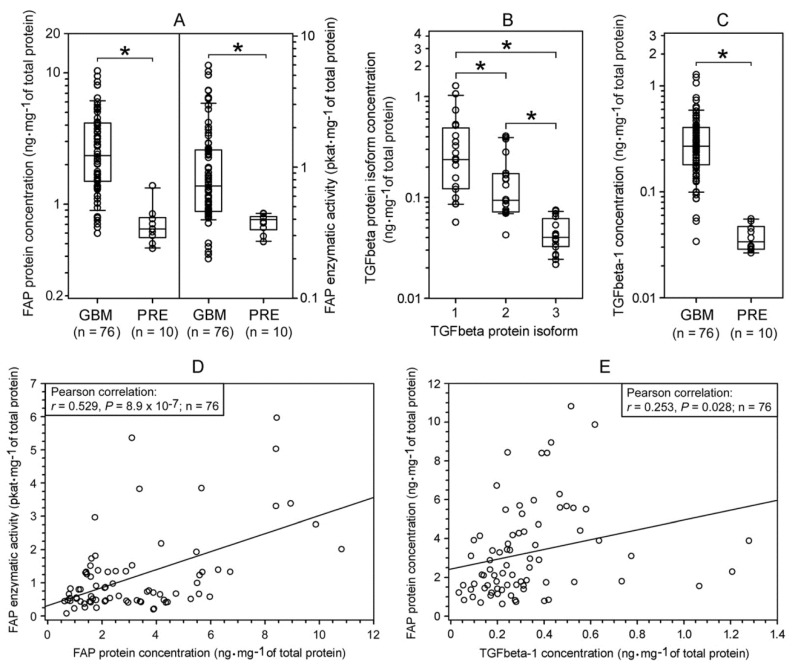
Expression of fibroblast activation protein (FAP) and transforming growth factor beta (TGFbeta) in human glioblastomas (GBMs). (**A**) Upregulation of FAP protein (left panel) and FAP enzymatic activity (right panel) levels in GBMs as compared to pharmacoresistant epilepsy (PRE) brain tissues. (**B**) Differential expression of TGFbeta protein isoforms in GBMs (n = 20). (**C**) Upregulation of TGFbeta-1 protein level in GBMs as compared to PRE brain tissues. (**D**) Relationship between the levels of FAP enzymatic activity and FAP protein in GBMs. (**E**) Relationship between FAP protein and TGFbeta-1 protein levels in GBMs. In (**A**–**C**), the sets of individual data points (each representing the mean of measurements in triplicate) are presented as a median with the box showing the 25th–75th percentile and whiskers indicating the 10th and 90th percentile. Note the logarithmic y-axis. In (**A**,**C**), * *p* < 0.01, Mann–Whitney rank sum test. In (**B**), * *p* < 0.05, Kruskal–Wallis one way ANOVA on ranks.

**Figure 2 ijms-22-01046-f002:**
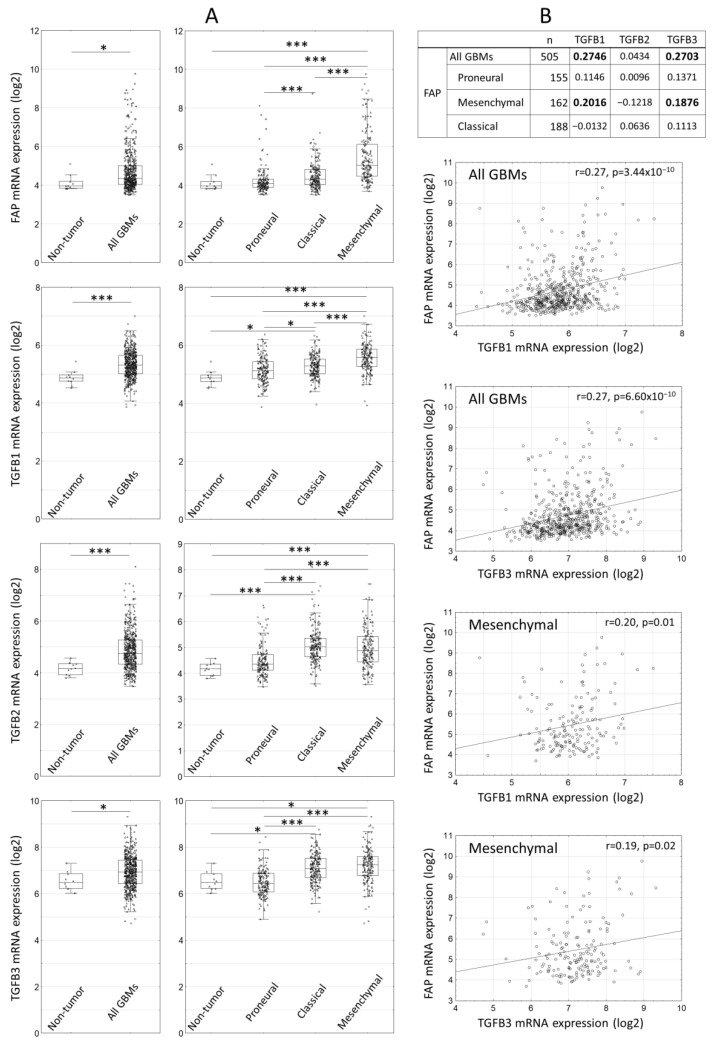
FAP and TGFbeta expression in molecular subtypes of primary glioblastomas (GBMs) according to The Cancer Genome Atlas (TCGA) data. (**A**) Expression of FAP and individual isoforms of TGFbeta mRNA in GBMs and individual molecular subtypes compared to control brain tissues. Line—median, box—25th–75th percentile, whiskers—non-outlier range, triangles—raw data, circles—outliers, asterisk—extremes; * *p* < 0.05, *** *p* < 0.001, Mann–Whitney U test and Kruskal–Wallis one way ANOVA on ranks. (**B**) Correlation between FAP and individual TGFbeta isoforms in all GBMs and in individual molecular subtypes of GBM (Pearson correlation coefficient, correlations with *p* < 0.05 are in bold). Scatter plots are shown only for statistically significant correlations.

**Figure 3 ijms-22-01046-f003:**
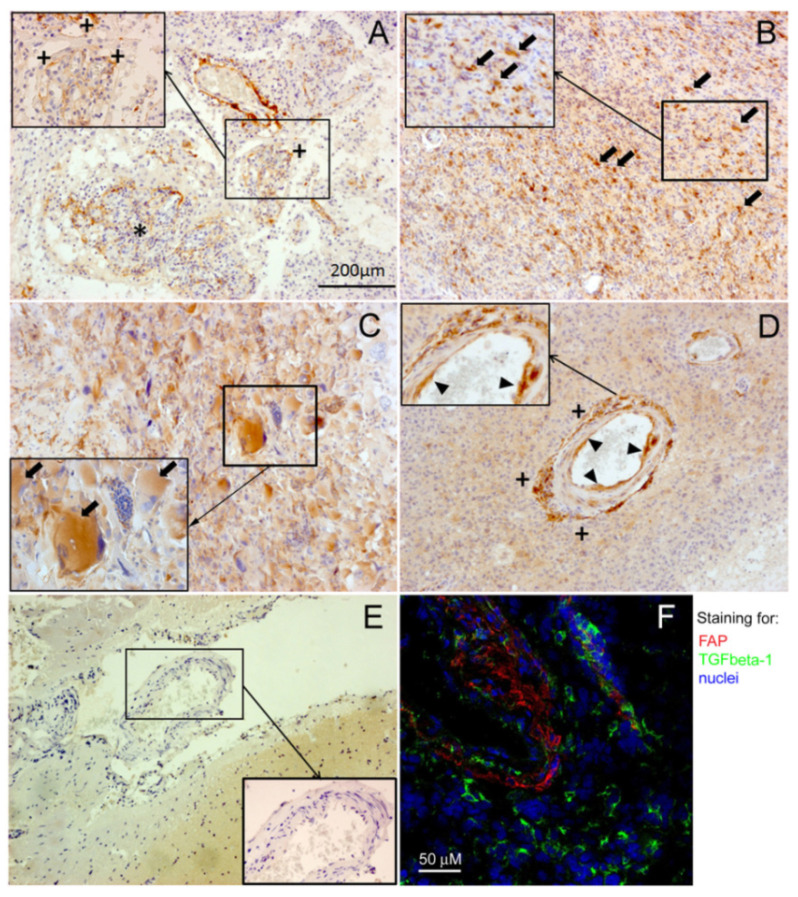
Immunohistochemical detection of FAP and TGFbeta-1 in the tumor microenvironment of human glioblastomas (GBMs). (**A**–**D**) Representative images of FAP immunopositivity (brown) in GBMs as compared to (**E**) non-tumorous brain tissue (pharmacoresistant epilepsy), where no FAP immunostaining was detected. In GBMs, FAP is expressed in perivascular stromal cells (+), including microvascular proliferations (*), and in some tumors also in some cancer cells (arrows). Infrequently, FAP was also detected in the endothelial cells of blood vessels (arrowheads showing endothelial FAP in an arteriole). (**F**) Representative confocal fluorescence microscopy image of a GBM tissue section showing TGFbeta-1 immunopositivity (green) in the tumor parenchyma and stromal regions in the vicinity of FAP expressing perivascular stromal cells (red). Nuclei (blue) were counterstained with the Hoechst 33258 dye.

**Figure 4 ijms-22-01046-f004:**
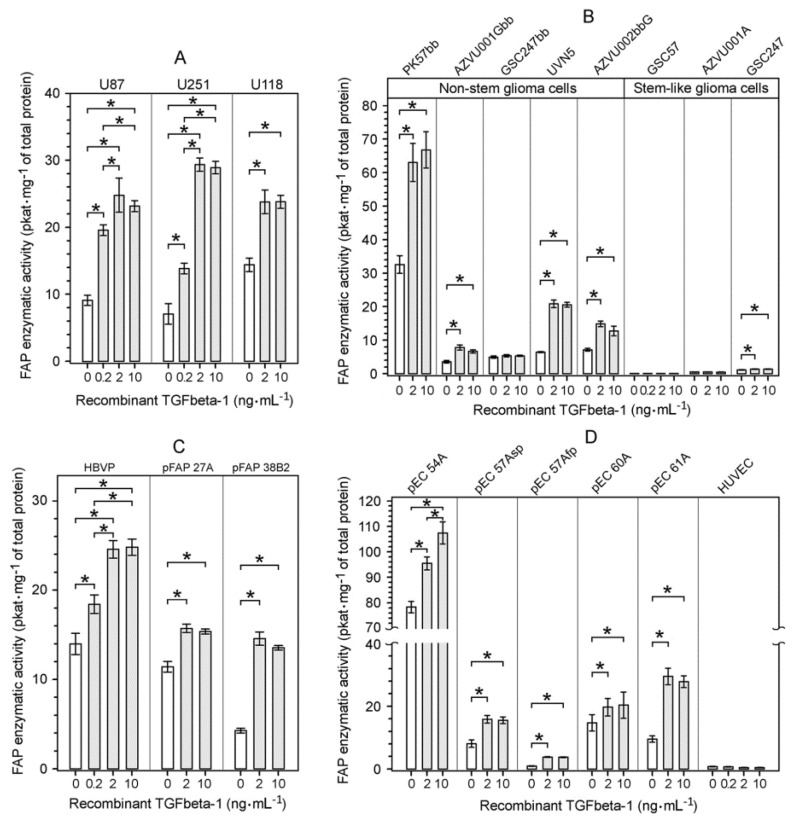
Upregulation of FAP enzymatic activity by human recombinant TGFbeta-1 protein in different cell types originating from human glioblastomas (GBMs). The cells were cultured in the absence or presence of recombinant TGFbeta-1 for 72 h and their lysates were assayed for FAP enzymatic activity and total protein concentration. (**A**) Upregulation of FAP enzymatic activity in TGFbeta-1-treated permanent glioma cell lines. (**B**) Upregulation of FAP enzymatic activity in TGFbeta-1-treated non-stem glioma cell cultures, but absence of or negligible increase in FAP enzymatic activity in TGFbeta-1-treated glioma stem-like cell cultures. (**C**) Upregulation of FAP enzymatic activity in TGFbeta-1-treated human brain vascular pericytes (HBVP) and glioblastoma-derived FAP^+^ mesenchymal (pFAP) cell cultures. (**D**) Upregulation of FAP enzymatic activity in TGFbeta-1-treated endothelial cell (pEC) cultures isolated from human GBMs, but not in human umbilical vein endothelial cells (HUVEC). Results are presented as mean ± SD from six parallel cell cultures measured in triplicate. In (**A**–**D**), * *p* < 0.05, one way ANOVA, Tukey’s post hoc test.

**Figure 5 ijms-22-01046-f005:**
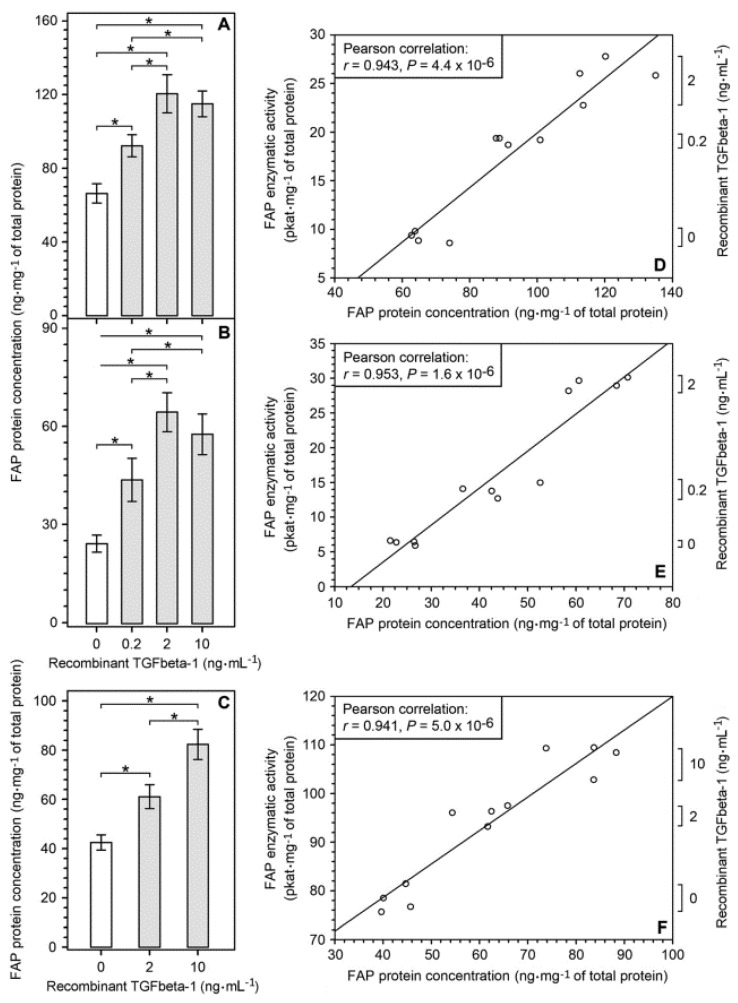
Upregulation of FAP protein concentration by human recombinant TGFbeta-1 protein in permanent glioma cell lines and endothelial cells (pEC) isolated from human glioblastoma. The cells were cultured without TGFbeta-1 and with several concentrations of recombinant TGFbeta-1 for 72 h and their lysates were assayed for FAP protein concentration, FAP enzymatic activity and total protein concentration. (**A**,**B**) Upregulation of FAP protein concentration in TGFbeta-1-treated glioma cell lines U87 (**A**) and U251 (**B**). (**C**) Upregulation of FAP protein concentration in TGFbeta-1-treated pEC 54A cells. (**D**,**E**) Relationship between FAP protein concentration and FAP enzymatic activity in U87 (**D**) and U251 (**E**) cells cultured without and with TGFbeta-1. (**F**) Relationship between FAP protein concentration and FAP enzymatic activity in pEC 54A cells cultured without and with TGFbeta-1. In **A**–**C**, results are presented as mean ± SD from four parallel cell cultures measured in triplicate. * *p* < 0.05, one way ANOVA, Tukey’s post hoc test. In (**D**–**F**), Pearson correlation between FAP protein concentration and FAP enzymatic activity in U87 (**D**), U251 (**E**) and pEC 54A (**F**) cells is shown.

**Figure 6 ijms-22-01046-f006:**
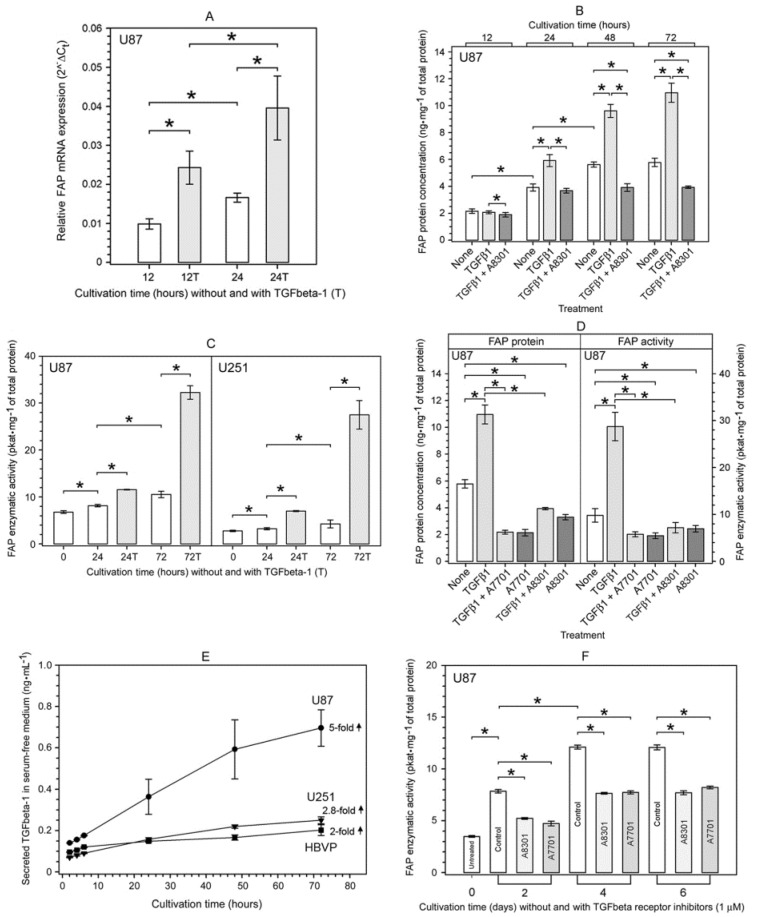
Characterization of TGFbeta-1-mediated upregulation of FAP mRNA, FAP protein, and FAP enzymatic activity in human glioma cells. (**A**) Time-dependent upregulation of FAP mRNA expression in U87 glioma cells during cultivation without and with 10 ng⋅mL^−1^ of human recombinant TGFbeta-1. (**B**) Time-dependent upregulation of FAP protein in U87 glioma cells during cultivation without and with 10 ng⋅mL^−1^ of human recombinant TGFbeta-1. A complete blocking of exogenous and partial blocking of endogenous FAP protein upregulation by co-treatment with the TGFbeta type I receptor inhibitor A8301 (1 µM). (**C**) Time-dependent upregulation of FAP enzymatic activity in U87 and U251 glioma cells during cultivation without and with 10 ng⋅mL^−1^ of human recombinant TGFbeta-1. (**D**) Upregulation of both FAP protein and FAP enzymatic activity in U87 glioma cells after cultivation with 10 ng⋅mL^−1^ of human recombinant TGFbeta-1 for 72 h and its complete blocking by co-treatment with the TGFbeta type I receptor inhibitors A7701 and A8301 (both at 1 µM), which also reduced the baseline FAP protein and FAP enzymatic activity levels. (**E**) Time course of secretion of endogenous TGFbeta-1 into serum-free media by cultured glioma cells U87 and U251 and human brain vascular pericytes (HBVP). The mean increase in secreted TGFbeta-1 level between the 2nd and 72nd hour of cultivation is indicated for each examined cell line. (**F**) Time course of increase in the baseline level of FAP enzymatic activity in cultured U87 glioma cells and its marked decrease by treatment with A8301 and A7701 (both at 1 µM). The results in (A) are presented as the mean ± SD from three parallel cell cultures measured in triplicate. Results in (**B**–**F**) are presented as the mean ± SD from five or six parallel cell cultures measured in triplicate. In A, * *p* < 0.05, one way repeated measures ANOVA. In (**B**,**C**,**F**), * *p* < 0.05, Friedman repeated measures ANOVA on ranks. In (**D**), * *p* < 0.05, one way ANOVA, Tukey’s post hoc test.

**Figure 7 ijms-22-01046-f007:**
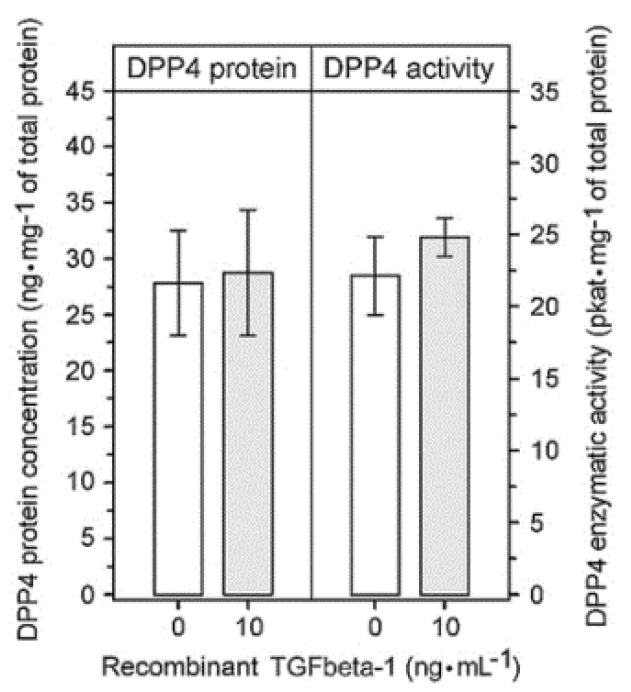
Effect of TGFbeta-1 on the expression of dipeptidyl peptidase-4 (DPP4) in human U87 glioma cells. The levels of DPP4 protein and DPP4 enzymatic activity were determined in U87 cell lysates after cultivating the cells with and without 10 ng⋅mL^−1^ of human recombinant TGFbeta-1 for 72 h. The results are presented as the mean ± SD from six parallel cell culture experiments for each condition measured in triplicate. The levels of DPP4 protein and DPP4 enzymatic activity in the TGFbeta-1-treated and TGFbeta-1-untreated cells did not significantly differ (*p* = 0.792 and *p* = 0.126, respectively; Mann–Whitney rank sum test).

**Figure 8 ijms-22-01046-f008:**
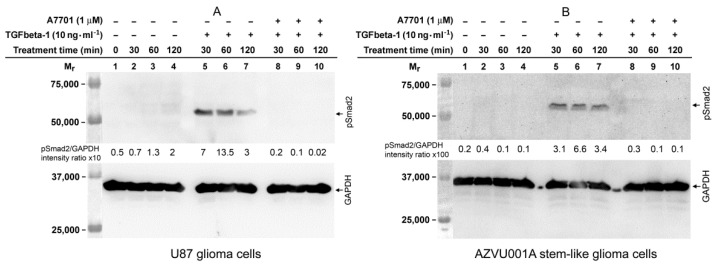
Western blot analysis of TGFbeta-1-induced phosphorylation of Smad2 protein in glioma cells. (**A**) U87 glioma cells and (**B**) AZVU001A glioma stem-like cells treated with recombinant TGFbeta-1 for the indicated times showed the presence of phosphorylated Smad2 (pSmad2) protein (lanes 5, 6, and 7, upper immunoblots). pSmad2 was undetectable in U87 glioma cells and in AZVU001A glioma stem-like cells cultured in the absence of TGFbeta-1 (lanes 2, 3, and 4, upper immunoblots) or in the presence of both TGFbeta-1 and TGFbeta type I receptor inhibitor A7701 (lanes 8, 9, and 10, upper immunoblots). The lower immunoblots show immunodetection of GAPDH which served as a protein loading control.

**Table 1 ijms-22-01046-t001:** Characteristics of patient cohorts.

Diagnosis	n	Age at Surgery ^a^ (Years)	Sex (Male/Female)	IDH ^b^ Status (Wild/Mutated)
Glioblastoma multiforme	76	62 (41–80)	49/27	76/0
Pharmacoresistant epilepsy	10	32 (22–62)	3/7	N.D. ^c^

^a^ Data are presented as the median with the range in parentheses; ^b^ IDH = isocitrate dehydrogenase; ^c^ N.D. = not determined.

**Table 2 ijms-22-01046-t002:** Antibodies used for the characterization of cell cultures derived from human glioblastomas.

Antibody, Source	Dilution	Temperature, Incubation Time	Secondary Antibody (1:500, Room Temperature, 1 h)
Anti-FAP mouse IgG1 kappa isolated from F19 mouse hybridoma (ATCC CRL-2733)	56 μg·mL^−1^	4 °C, overnight	Anti-mouse Alexa Fluor 488 (Invitrogen, A21202)
Anti-TE-7 (CBL271, Millipore)	1:100	room temperature, 1 h	Anti-mouse Alexa Fluor 488 (Invitrogen, A21202)
Anti-PDGFRbeta (LS-C11443, LSBio)	1:50	4 °C, overnight	Anti-mouse Alexa Fluor 488 (Invitrogen, A21202)
Anti-vWF (A0083, Dako)	1:200	room temperature, 1 h	Anti-rabbit Alexa Fluor 488 (Invitrogen, A11010)
Anti-GFAP (11-255-M001, Exbio)	1:200	4 °C, overnight	Anti-mouse Alexa Fluor 488 (Invitrogen, A21202)
Anti-beta III Tubulin (TU-20, ab7751, Abcam)	1:250	4 °C, overnight	Anti-mouse Alexa Fluor 488 (Invitrogen, A21202)

**Table 3 ijms-22-01046-t003:** Primers and TaqMan probes for real-time RT-PCR quantification of expression of human FAP and ACTB mRNAs.

Human Transcript	GenBank Accession No.	Primer Names	Sequences of Primers and TaqMan Probes
Fibroblast activation protein (FAP)	NM_004460.4	Forward primer:	5′-TCTGCTGTGCTTGCCTTATTG-3′
		Reverse primer:	5′-ATGAAGATATTCTTGTCCTGAAATCC-3′
		TaqMan probe:	5′-(6-FAM)TGCATTGTCTTACGCCCTTCAAGAGTTCA(TAMRA)-3′
β-actin (ACTB)	NM_001101.2	Forward primer:	5′-ATGGCCACGGCTGCTT-3′
		Reverse primer:	5′-CCATGCCCAGGAAGGAA-3′
		TaqMan probe:	5′-(6-FAM)CCCTGGAGAAGAGCTACGAGCTGCCT(TAMRA)-3′

## Data Availability

The datasets generated during and/or analysed during the current study are available from the corresponding author on reasonable request. Publicly available datasets from the TCGA were downloaded from http://gliovis.bioinfo.cnio.es.

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
