# Peer review of "Regulation of Fibroblast Activation Protein by Transforming Growth Factor Beta-1 in Glioblastoma Microenvironment"

_ijms, 2021, doi:10.3390/ijms22031046_

Round 1

Reviewer 1 Report

The manuscript ijms-1035850, Regulation of Fibroblast Activation Protein by Transforming Growth Factor Beta-1 in Glioblastoma Microenvironment presents a very well done research focused on FAP expression in glioblastomas and its correlation with TGFbeta. The manuscript is very well written and the graphical representations are very good. I did not find significant problems that need correction, but I would suggest a small conclusion section for the readers to better understand the whole data presented.

Author Response

As suggested by the reviewer, we included a small conclusion section for the readers to better understand the whole data presented (line numbers 594 – 599 in the revised MS with highlighted revisions).

Reviewer 2 Report

This is a well-written manuscript characterizing the cell type specific regulation of FAP by TGFB1 in glioblastoma.  The study shows that TGFB1 promotes the expression of FAP in non-stem glioma cells, pericytes, tumor endothelial and mesenchymal subpopulations (FAP+) but not in glioma stem cells.  This is linked to signaling through TGFB1 type I receptor and SMAD2.  Further, the authors show that TGFB1 is secreted by glioma cells to provide an autocrine mechanism for expression of FAP within the glioblastoma microenvironment.  In particular the differential response of the cells within the glioblastoma microenvironment is interesting and novel.

This manuscript is potentially of sufficient novelty and interest to the readership of IJMS to be published in this journal.  However, there are several deficiencies that need to be addressed and these are described below. 

  1. The evidence for a cell context specific role of TGFB in regulating expression of FAP, a potential GB therapeutic target, is well-documented and a strength of this manuscript.
  2. The evidence for a role of TGFB type I receptor and SMAD2 in regulating FAP expression is weak and needs to be strengthened. It is recommended that the authors utilize their primary cultures of the different cell types to address this deficiency by assessing the consequence of blocking TGFB type I receptor and SMAD2 on expression of FAP in these different cell types. 
  3. The methods description of the isolation and culture of the GB cell types is not sufficiently detailed to allow reviewers to determine the purity etc of these cultures. Were markers used to identify the various cell types?  If not these should be included to verify the various cell phenotypes. 
  4. Minor editing of the manuscript is recommended.

Author Response

1) The evidence for a cell context specific role of TGFB in regulating expression of FAP, a potential GB therapeutic target, is well-documented and a strength of this manuscript.

 We are glad for this opinion expressed by the reviewer.

2) The evidence for a role of TGFB type I receptor and SMAD2 in regulating FAP expression is weak and needs  to be strengthened. It is recommended that the authors utilize their primary cultures of the different cell types to address this deficiency by assessing the consequence of blocking TGFB type I receptor and SMAD2 on  expression of FAP in these different cell types.

Previous study in melanoma cell lines has demonstrated that TGFbeta-1 signaling through the TGFbeta receptor I leads to activation of Smad complexes, their binding to a specific SMAD-binding site in the FAP promoter and transcriptional stimulation of FAP expression ((Tulley and Chen 2014), see discussion line numbers 345-348 in the revised MS with highlighted revisions, reference 30). We therefore analyzed whether a similar mechanism may be involved in TGFbeta1-mediated upregulation of FAP in glioblastoma. Using glioma cells as a model, we obtained evidence for a role of TGFB type I receptor and SMAD signaling in regulating FAP expression with two TGFB type I receptor kinase inhibitors, Western blot analysis of TGFbeta-1-induced presence of phospho-SMAD2 and quantification of FAP mRNA. Our data are in line with the detailed work of Tulley et al. in a different cell type (Tulley and Chen 2014). We did not replicate our investigations in other studied cells, since we observed highly similar pattern of dose- a time-dependent response of FAP expression to the TGFbeta-1 treatment among other studied cell types including permanent glioma cell lines and primary cultures of non-stem glioma cells, pericytes, endothelial cells and pFAP+ mesenchymal cells. This suggests that TGFB type I receptor and SMAD signaling will play a similar role in regulating FAP expression in responsive cells.

We have modified the statement in abstract to more clearly indicate that our analysis was performed in glioma cells as a model to analyze the mechanism of TGFbeta1-mediated FAP upregulation in glioblastoma (line numbers 28-32).

3) The methods description of the isolation and culture of the GB cell types is not sufficiently detailed to allow reviewers to determine the purity etc of these cultures. Were markers used to identify the various cell types? If not these should be included to verify the various cell phenotypes.

The Material and Methods section 4.4. was substantially extended. More information regarding the isolation and characterization of the cell cultures is provided (lines 413-465). We also added one more table (Table 2 in the revised manuscript) which lists antibodies used for the detection of specific markers (line 464).  

4) Minor editing of the manuscript is recommended.

The manuscript was edited and small stylistic and typographical corrections were made throughout the text.

Round 2

Reviewer 2 Report

The authors have adressed my concerns adequately